# Impact of the COVID-19 pandemic on excess maternal deaths in Brazil: A two-year assessment

Jesem Douglas Yamall Orellana[1], Daniel Gray Paschoal Leventhal[2]*, María del Pilar Flores-Quispe[3], Lihsieh Marrero[4], Nadège Jacques[5], Lina Sofía Morón-Duarte[6,7], Cynthia Boschi-Pinto[8]

**1** Leônidas and Maria Deane Institute, Oswaldo Cruz Foundation, Manaus, Amazonas, Brazil, **2** VCU School of Medicine Medical Class of 2027, Richmond, Virginia, United States of America, **3** Center for Data and Knowledge Integration for Health (CIDACS), Gonçalo Moniz Institute, Oswaldo Cruz Foundation, Salvador, Bahia, Brazil, **4** Department of Nursing, Amazonas State University, Manaus, Amazonas, Brazil, **5** Center for Haitian Studies, Miami, Florida, United States of America, **6** Global Institute of Clinical Excellence, Keralty, Bogotá, Distrito Capital, Colombia, **7** Translational Research Group, Sanitas University Foundation, Bogotá, Distrito Capital, Colombia, **8** Department of Epidemiology and Biostatistics, Postgraduate Program in Collective Health, Institute of Collective Health, Fluminense Federal University, Niterói, Rio de Janeiro, Brazil

* leventhaldg@vcu.edu

**Data Availability Statement:** Data were obtained from the Mortality Information System (SIM) of the Brazilian Ministry of Health for the period of 2015-

## Abstract

### Background

Accurate estimates of the COVID-19 pandemic's indirect impacts are crucial, especially in low- and middle-income countries. This study aims to update estimates of excess maternal deaths in Brazil during the first two years of the COVID-19 pandemic.

### Methods

This was an exploratory mixed ecological study using the counterfactual approach. The observed maternal deaths were gathered from the Mortality Information System (SIM) for the period between March 2015 and February 2022. Expected deaths from March 2020 to February 2022 were estimated using quasipoisson generalized additive models, considering quadrimester, age group, and their interaction as predictor variables. Analyses were performed in R version 4.1.2, RStudio, version 2023.03.1+446 and carried out with support from the "mgcv" and "plot_model" libraries.

### Results

A total of 5,040 maternal deaths were reported, with varying excess mortality across regions and age groups, resulting in 69% excess maternal mortality throughout Brazil during the first two years of the pandemic. The Southeast region had 50% excess mortality throughout the first two years and 76% excess in the second year. The North region had 69% excess mortality, increasing in the second year, particularly among women aged 20–34. The Northeast region showed 80% excess mortality, with a significant increase in the second year, especially among women aged 35–49. The Central-West region had 75% excess mortality, higher in the second year and statistically significant among women aged 35–49. The South

2022 (https://datasus.saude.gov.br/transferencia-de-arquivos/).

**Funding:** YES - JDYO received funding from PROEP-LABS/ILMD/FIOCRUZ AMAZÔNIA (2023-2025). The funders played no role in the study design, analysis, and manuscript preparation.

**Competing interests:** The authors have declared that no competing interests exist.

region showed 117% excess mortality, reaching 203% in the second year among women aged 20–34, but no excess mortality in the 10–19 age category.

## Conclusions

Over two years, Brazil saw a significant impact on maternal excess deaths, regardless of region and pandemic year. The highest peak occurred between March and June 2021, emphasizing the importance of timely and effective epidemic responses to prevent avoidable deaths and prepare for new crises.

## Introduction

Excess mortality due to the COVID-19 pandemic has been widely documented for several causes of death [1–4]. The World Health Organization (WHO) recently estimated that 14.8 million excess deaths have occurred globally as a result; Brazil ranked fifth out of 25 countries with the highest estimated excess mortality [1]. COVID-19 can directly cause death, yet the pandemic's indirect impact, including the disruption of healthcare systems, limited access to health services and reduced availability of health care professionals and resources, is also of major concern.

In June 2023, cumulative COVID-19 mortality in Brazil reached 3,265 per million inhabitants (approximately four times the world average [5]), a disaster that has affected the health and living conditions of millions of Brazilians. The tragedy has also been reflected in the increased risk of maternal mortality in the country. An analysis by the Brazilian Ministry of Health estimated the country's maternal mortality ratio (MMR) to be consistently around 60 deaths for every 100,000 live births from 2015 to 2019, far from the goal of a maximum of 30 maternal deaths for every 100,000 live births by 2030, and the 2020 MMR (74.7/100,000) was 29% larger than that of the previous year (57.9/100,000) [6]. In 2021, average weekly deaths in pregnant and puerperal women increased by 151%, more than double the increase in the general population [7].

A previous study examining data from March 2020 to May 2021 estimated a 70% excess in maternal mortality in Brazil, with wide variations by region and time period evaluated [2]. However, the study did not allow for an evaluation of the possible indirect effects of the COVID-19 epidemic on maternal mortality in traditional age groupings (10–19 years, 20–34 years, and 35–49 years). In addition, the study did not assess excess maternal deaths in the second and most critical year of the epidemic in different regions of Brazil, especially in the months following the late, uninterrupted distribution of vaccines against COVID-19 to pregnant and post-partum women beginning in July 2021.

The higher mortality observed in pregnant women in Brazil as compared to other countries is likely due to its chronic structural problems with women's health care, including difficulties accessing care and low-quality prenatal care, among others [8]. It is important to note that, even when COVID-19 is directly involved in the deaths of these women, failures in the health system and medical care also likely contributed to the fatal outcome. Of all maternal deaths identified in Brazil in 2020, 28% did not have access to an intensive care unit (ICU) bed and 36% were not intubated nor received mechanical ventilation [9].

Given this scenario, this study aimed to update previous estimates of excess maternal deaths in Brazil during the first two years of the COVID-19 pandemic, using the counterfactual approach.

## Methods

### Study type, data source, and units of analysis

This was an exploratory mixed ecological study [10] using the counterfactual approach [11]. Data were obtained from the Mortality Information System (SIM) of the Brazilian Ministry of Health for the period of 2015 to 2022. Given Brazil's vast regional inequities in COVID-19 outcomes [12, 13], the country's five macroregions (North, Northeast, Central-West, South, Southeast) were employed as units of analysis. The North and Northeast regions of the country are historically socioeconomically disadvantaged as compared to Brazil's Southeast and South, the centers of Brazil's industrial capacity and urbanization [14].

Data from 2015 to 2021 [15] are considered finalized. In order to ensure reliability of the 2022 data, only those deaths occurring up to February 2022 were included.

### Working definitions

We selected maternal deaths in accordance with the criteria of the Brazilian Ministry of Health's protocol of special codes for mortality. This includes maternal mortality in women aged 10–49 years and considers the victim's place of residence. Maternal death is death occurring at any point between the beginning of gestation and the 42$^{nd}$ day after the termination of gestation, irrespective of the location or duration of gestation and due to any cause related to or exacerbated by the gestation or related measures, except for accidental or incidental causes [16].

Our inclusion criteria for maternal deaths followed codings from the International Statistical Classification of Diseases and Related Health Problems, 10$^{th}$ Revision, as shown in Table 1.

In the event of inconsistencies between the declared cause of maternal death and the time of death (during the pregnancy, delivery, or miscarriage/abortion, during the postpartum period up to 42 days, during the postpartum period, between 43 days and 1 year or outside these periods), information on the basic cause of death was prioritized [16, 17]. Excess deaths represent the number of deaths above an expected value following a previously observed mortality pattern in the population, according to the World Health Organization [18].

### Data analysis

We analyzed data from March 2020 to February 2022. We did not apply correction factors to the maternal mortality estimates, as the focus of the article is to present and discuss excess

**Table 1. Maternal deaths according to codings from the International Statistical Classification of Diseases and Related Health Problems, 10$^{th}$ Revision (ICD-10).**

| ICD-10 Codings | Group of maternal deaths |
|---|---|
| O00-O99 | Pregnancy, childbirth and the puerperium* |
| B20-B24 | Human immunodeficiency virus [HIV] disease** |
| D39.2 | malignant or invasive hydatidiform mole** |
| E23.0 | Hypopituitarism** |
| M83.0 | Puerperal osteomalacia*** |
| A34 | Obstetrical tetanus*** |
| F53 | Mental and behavioural disorders associated with the puerperium, not elsewhere classified*** |

*Except deaths outside of the 42-day postpartum period (codes O96 and O97)

**With a gestational report at the time of death or up to 42 days before death

***With death occurring up to 42 days after the termination of pregnancy or when there was no information on the time elapsed between the termination of gestation and death

deaths as a percentage. These do not change, with or without a fixed correction on the observed or estimated counts. The study period was divided into six four-month periods (quadrimesters): March to June 2020, July to October 2020, November 2020 to February 2021, March to June 2021, July to October 2021, and November 2021 to February 2022. The chosen level of aggregation (by quadrimester) was agreed upon with the aim to improve the accuracy of our predictions, especially those pertaining to maternal deaths in women aged 10–19 years or even in subanalyses in regions with smaller samples, like the Central-West and South regions. The observed deaths in SIM from March 2015 to February 2020 were used to calculate the expected deaths in each of the six quadrimesters assessed.

Studies have shown that the counterfactual approach to evaluating excess mortality is useful for estimating the indirect effects of crises that severely compromise healthcare services and in which socioeconomically marginalized groups suffer the most deaths, as has occurred during the COVID-19 pandemic. The approach is a reasonable representation of the difference between observed deaths and the deaths that would be expected in the absence of the pandemic [11]. We obtained estimates of expected deaths from March 2020 to February 2022 through *quasipoisson* generalized additive models, adjusted for overdispersion [19]. The variables quadrimester, age group, and the interaction between age group and year of occurrence were considered predictors in each regional model and at the national level, with p-values <20% considered statistically significant. We adjusted the year of occurrence non-parametrically (spline) to capture possible non-linear mortality trends in the study period. We estimated the number of expected deaths with the adjusted models. Age groups were aggregated in the following brackets: 10–19 years, 20–34 years, and 35–49 years. Building on this approach, in our study, we adopted a Generalized Additive Model (GAM) with a quasipoisson approach:

$$Y|X \sim Poisson(\theta)$$
$$Log(\theta) = \beta_0 + f(year|age) + \beta_1 \times age + \beta_2 \times quadrimesters$$

where θ represents the mean count and y follows a *Poisson* distribution with a mean parameter θ. The mean parameter θ specifically is used to address overdispersion (when the variance of count data is larger than the mean) and does not have a specific associated distribution, acting as a scaling factor that adjusts the variance to reflect the observed overdispersion. Therefore, the modeled distribution is proportional to the mean estimated by the model, expressed mathematically as $Var(Y|X) = ø \cdot µ$, where µ is the expected mean. Furthermore, when the value of ø is larger than one, this indicates that the data display more variability than expected (assuming a *Poisson* distribution in which both the mean and the variance are equal). The function f (year|age) captures non-linear relationships between 'year' and y, adjusted for 'age'. This model not only provides flexibility in examining the relationships among variables but also efficiently accommodates the inherent variability in the data through the *quasipoisson* adjustment, making it an insightful choice for analyzing the trends and patterns in our study.

We based our estimates of excess maternal mortality on the calculated ratio between the number of observed maternal deaths and those expected if the pandemic did not occur for each macroregion and at the country level, stratified by age group and quadrimester [11]. We compared the width of the 95% confidence intervals of each of the expected point estimates to the corresponding observed value in order to assess statistically significant differences between the observed and expected values. The results of excess deaths were expressed in percentages, which, per the WHO, tend to be more understandable to decision-makers [18]. We performed our analyses in R version 4.1.2, RStudio, version 2023.03.1+446 with support from the "mgcv" and "plot_model" libraries (https://www.r-project.org).

## Ethical considerations

This study did not need approval from a human research ethics committee as the data used were fully anonymized and publicly available, in accordance with Resolution No. 510/2016 of the National Health Council. Data were accessed on March 15, 2023.

## Results

Between March 2020 and February 2022, 5,040 maternal deaths were reported in Brazil, with 1,739 in the Southeast, 1,516 in the Northeast, 743 in the North, 558 in the South, and 484 in the Central-West. Throughout Brazil, during the first two years of the pandemic, there was a 69% (5,040/2,986) excess in maternal mortality, reaching 100% (2,935/1,468) during the second year, and 39% (2,104/1,518) in the first. There was also excess maternal mortality regardless of age group in the second year (Table 2). In the 20-34-year-old age group there was excess maternal mortality regardless of quadrimester, and from March to June 2021 there was an excess in maternal deaths regardless of age group (Fig 1).

In the Southeast region, a 50% (1,739/1,162) excess in maternal mortality was observed in the first two years of the pandemic, with 73% (1,117/646) excess in the 20–34 years age group. There was a larger excess in maternal deaths during the second year of the pandemic (76%

**Table 2. Observed and expected maternal deaths, Brazil and Southeast region, 2020–2022.**

| Characteristics | Observed (n) | Expected (n) | Expected* (95CI%) | Ratio |
|---|---|---|---|---|
| **Brazil** | | | | |
| Age Group (years) | | | | |
| 10–19 | 161 | 190 | **161–220** | **0.84** |
| 20–34 | 1290 | 900 | **832–969** | **1.43** |
| 35–49 | 654 | 428 | **362–495** | **1.53** |
| Year 1 | 2105 | 1518 | **1355–1684** | **1.39** |
| 10–19 | 230 | 185 | **149–221** | **1.24** |
| 20–34 | 1852 | 871 | **789–953** | **2.13** |
| 35–49 | 853 | 412 | **319–505** | **2.07** |
| Year 2 | 2935 | 1468 | **1257–1679** | **2.00** |
| 10–19 | 391 | 375 | 310–441 | 1.04 |
| 20–34 | 3142 | 1771 | **1621–1922** | **1.77** |
| 35–49 | 1507 | 840 | **681–1000** | **1.79** |
| Year 1 and 2 | 5040 | 2986 | **2612–3363** | **1.69** |
| **Southeast** | | | | |
| Age Group (years) | | | | |
| 10–19 | 44 | 61 | 41–79 | 0.72 |
| 20–34 | 462 | 328 | **282–375** | **1.41** |
| 35–49 | 220 | 196 | 160–233 | 1.12 |
| Year 1 | 726 | 585 | **483–687** | **1.24** |
| 10–19 | 63 | 60 | 37–83 | 1.05 |
| 20–34 | 655 | 318 | **264–374** | **2.06** |
| 35–49 | 295 | 199 | **153–244** | **1.48** |
| Year 2 | 1013 | 577 | **454–701** | **1.76** |
| 10–19 | 107 | 121 | 78–162 | 0.88 |
| 20–34 | 1117 | 646 | **546–749** | **1.73** |
| 35–49 | 515 | 395 | **313–477** | **1.30** |
| Year 1 and 2 | 1739 | 1162 | **937–1388** | **1.50** |

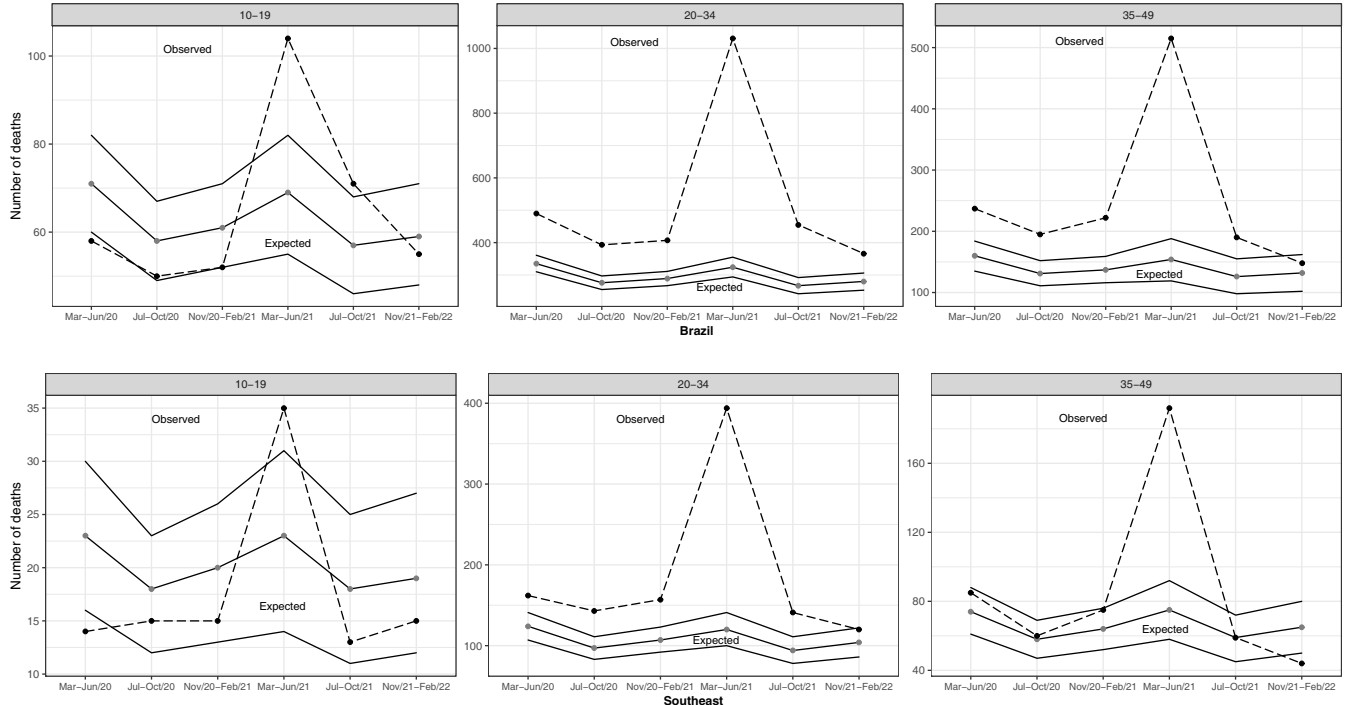

**Fig 1. Observed and expected maternal deaths according to age group and six consecutive four-month periods, Brazil and Southeast region, 2020 to 2022.**

[1,013/577]) than the first. From March to June 2021 there was excess maternal mortality in all age groups (Fig 1), while a 32% drop (65 expected [95% Confidence Interval– 95%CI: 50–80], 44 observed) occurred among women aged 35–49 from November 2021 to February 2022 (Fig 1).

In the North region, a 69% excess in maternal deaths was observed in the first two years of the pandemic. Excess maternal mortality was greater in the second year (83% [395/216]) than in the first, especially among women 20–34 years of age (117% [260/120]) (Table 3). In the 35–49 age group, excess mortality occurred in all quadrimesters, except November 2021 to February 2022. There was no excess mortality in the 10–19 age group, regardless of quadrimester (Fig 2).

In the Northeast, 80% excess maternal mortality was observed during the first two years of the pandemic, especially in women 35–49 years old (113% [469/220]). Excess maternal deaths were also greater in the second year, at 107% (839/405), and 147% (257/104) among women aged 35–49 (Table 3). Excess maternal mortality was observed regardless of quadrimester in both the 20–34 and 35–49 age groups, in addition to excess maternal deaths in the 10–19 age range in two out of six quadrimesters evaluated (Fig 2).

In the Central-West region, a 75% (484/276) excess in maternal mortality was observed during the pandemic's first two years, with 138% (152/64) excess among women aged 35–49. Excess maternal mortality was also larger in the second (123% [306/137]) than in the first pandemic year (28% [178/139]), with a 181% (87/31) excess in women 35–49 years old (Table 4). Excess maternal deaths occurred from March to June 2021 regardless of age group, and in women aged 35–49, maternal deaths occurred in all quadrimesters except November 2021 to February 2022 (Fig 3).

**Table 3. Observed and expected maternal deaths, North and Northeast regions, 2020–2022.**

| Characteristics | Observed (n) | Expected (n) | Expected* (95CI%) | Ratio |
|---|---|---|---|---|
| **North** | | | | |
| Age Group (years) | | | | |
| 10–19 | 37 | 45 | 32–57 | 0.82 |
| 20–34 | 213 | 128 | **96–159** | **1.66** |
| 35–49 | 98 | 50 | **37–63** | **1.96** |
| Year 1 | 348 | 223 | **165–279** | **1.56** |
| 10–19 | 48 | 45 | 29–62 | 1.07 |
| 20–34 | 260 | 120 | **78–162** | **2.17** |
| 35–49 | 87 | 51 | **34–69** | **1.71** |
| Year 2 | 395 | 216 | **141–293** | **1.83** |
| 10–19 | 85 | 90 | 61–119 | 0.94 |
| 20–34 | 473 | 248 | **174–321** | **1.91** |
| 35–49 | 185 | 101 | **71–132** | **1.83** |
| Year 1 and 2 | 743 | 439 | **306–575** | **1.69** |
| **Northeast** | | | | |
| Age Group (years) | | | | |
| 10–19 | 59 | 55 | 42–71 | 1.07 |
| 20–34 | 406 | 265 | **229–302** | **1.53** |
| 35–49 | 212 | 116 | **81–150** | **1.83** |
| Year 1 | 677 | 436 | **352–523** | **1.55** |
| 10–19 | 78 | 52 | **34–68** | **1.50** |
| 20–34 | 504 | 249 | **207–291** | **2.02** |
| 35–49 | 257 | 104 | **58–150** | **2.47** |
| Year 2 | 839 | 405 | **299–509** | **2.07** |
| 10–19 | 137 | 107 | 76–139 | 1.28 |
| 20–34 | 910 | 514 | **436–593** | **1.77** |
| 35–49 | 469 | 220 | **139–300** | **2.13** |
| Year 1 and 2 | 1516 | 841 | **651–1032** | **1.80** |

In the South region, there was 117% (558/257) excess in maternal deaths throughout the first two years of the pandemic, reaching 203% (382/126) in the second year. There were no excess maternal deaths in the 10–19 age category (Table 4). Excess maternal mortality was observed from June 2020 to February 2022 in women aged 20–34, and from June 2020 to October 2021 in women aged 35–49, reaching 413% (77/15) in March to June 2021, and with an equal number of observed and expected deaths (without statistical significance) from November 2021 to February 2022 (Fig 3).

The interaction analysis was statistically significant ($p < 0.20$) in at least one age category in all macroregions except the Central-West. The analysis was statistically significant ($p < 0.20$) in all age categories at the national level, and in the Northeast region (Table 5).

Roughly 87% of all maternal deaths were due to (1) complications related to the puerperium/other obstetric conditions, (2) oedema, proteinuria and hypertensive disorders in pregnancy, childbirth, and the puerperium, and (3) other complications of labour and delivery (S1 Table). There were consistent patterns of maternal deaths due to causes grouped in these three blocks across quadrimesters, although the largest number of deaths were due to other viral diseases complicating pregnancy, childbirth and the puerperium, especially from March to June 2021 (S2 Table).

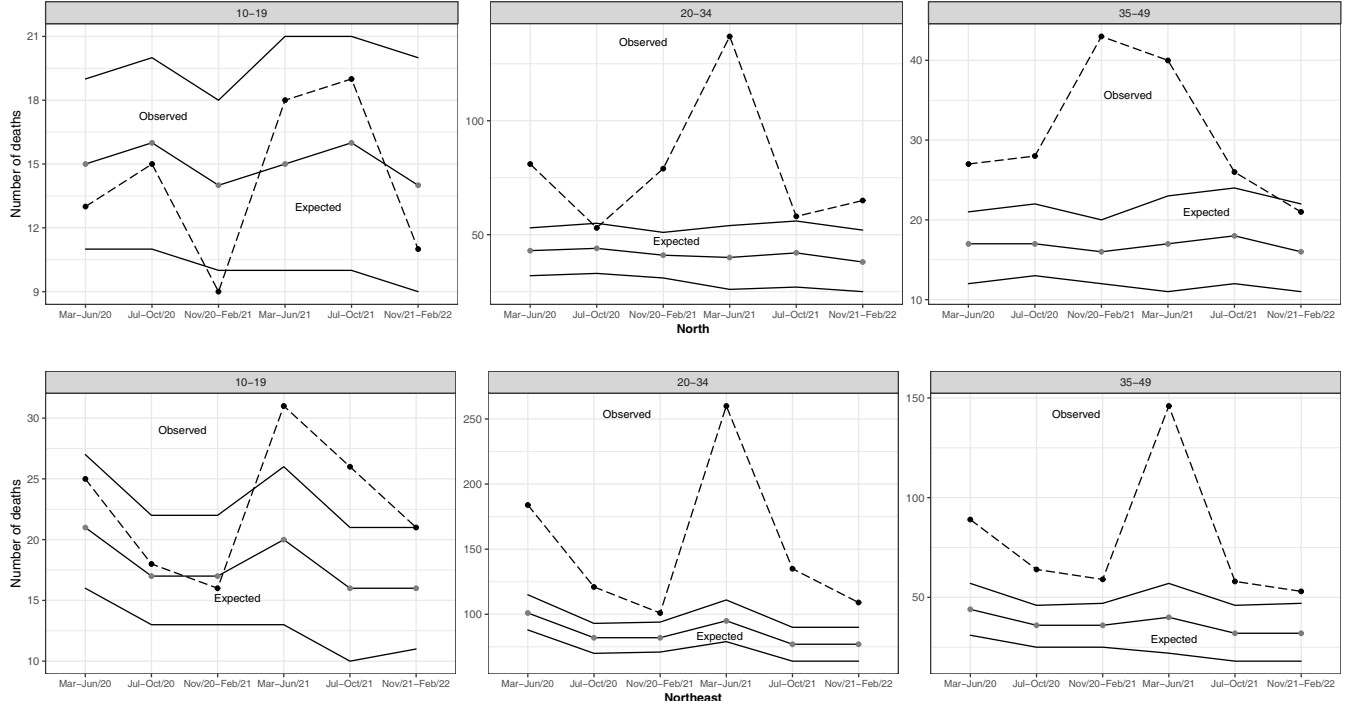

**Fig 2. Observed and expected maternal deaths according to age group and six consecutive four-month periods, North and Northeast regions, 2020 to 2022.**

## Discussion

The impact of the COVID-19 pandemic on excess maternal deaths in Brazil was extremely high throughout the first two years, with 100% excess in the second year. Although the North and Northeast were the two most impacted regions in the first year, excess maternal mortality was markedly higher in the Central-West and South regions during the second year, reaching 413% in women aged 35–49 from the South in March to June 2021 (the worst quadrimester of all, regardless of region).

Our national estimates did not show a significant difference between observed and expected maternal deaths in the 10–19 age group for the entire study period (375 expected [95%CI: 310–441], 391 observed). However, there was 24% excess during the second year (reaching 50.8% [104/69] in March to June/2021). There was also substantial variation in annual excess maternal deaths at the regional level, with values ranging from 5% to 54% during the second year, despite this change being significant only in the Northeast region (one of the poorest regions of the country). Finally, despite uncertainty in some estimates, probably due to the small number of deaths among women and girls aged 10–19 years, there were substantial differences in excess maternal deaths between years and regions, reflecting both the worsening of the pandemic situation in Brazil over time and regional inequalities [20, 21].

The 20–34-year age group was one of the most affected, showing a significant change between observed and expected maternal deaths at the national level. The greatest excess of maternal deaths occurred between November 2020 and October 2021, with 113% excess in the second year overall. The same pattern was observed regionally with a second-year increase of 102% in the Northeast, 106% in the Southeast, 117% in the North, 125% in the Midwest, and 231% in the South.

**Table 4. Observed and expected maternal deaths, Central-West and South regions, 2020–2022.**

| Characteristics | Observed (n) | Expected (n) | Expected* (95CI%) | Ratio |
|---|---|---|---|---|
| **Central-West** | | | | |
| Age Group (years) | | | | |
| 10–19 | 11 | 18 | 10–25 | 0.61 |
| 20–34 | 102 | 88 | 70–109 | 1.16 |
| 35–49 | 65 | 33 | **22–43** | **1.97** |
| Year 1 | 178 | 139 | **102–177** | **1.28** |
| 10–19 | 21 | 18 | 7–28 | 1.17 |
| 20–34 | 198 | 88 | **64–113** | **2.25** |
| 35–49 | 87 | 31 | **18–45** | **2.81** |
| Year 2 | 306 | 137 | **89–186** | **2.23** |
| 10–19 | 32 | 36 | 17–53 | 0.89 |
| 20–34 | 300 | 176 | **134–222** | **1.70** |
| 35–49 | 152 | 64 | **40–88** | **2.38** |
| Year 1 and 2 | 484 | 276 | **191–363** | **1.75** |
| **South** | | | | |
| Age Group (years) | | | | |
| 10–19 | 10 | 13 | 7–20 | 0.77 |
| 20–34 | 107 | 75 | **56–94** | **1.43** |
| 35–49 | 59 | 43 | **30–55** | **1.37** |
| Year 1 | 176 | 131 | **93–169** | **1.34** |
| 10–19 | 20 | 13 | 6–22 | 1.54 |
| 20–34 | 235 | 71 | **48–96** | **3.31** |
| 35–49 | 127 | 42 | **27–58** | **3.02** |
| Year 2 | 382 | 126 | **81–176** | **3.03** |
| 10–19 | 30 | 26 | 13–42 | 1.15 |
| 20–34 | 342 | 146 | **104–190** | **2.34** |
| 35–49 | 186 | 85 | **57–113** | **2.19** |
| Year 1 and 2 | 558 | 257 | **174–345** | **2.17** |

These substantial excesses in maternal mortality were closely linked to the public health system's immense difficulties in caring for pregnant and postpartum women during the pandemic. There were many barriers to accessing obstetric services, such as racial inequities in the delivery of maternal care services, poor-quality prenatal care, insufficient resources to manage emergency and critical care, and the overall collapse of the health system during the pandemic period [9, 22, 23].

The 35-49-year-old age group was also severely impacted by excess maternal mortality during the first two years of the epidemic in the country, with significant excess mortality in years one and two regardless of region, except in the Southeast during the first year. The North region was the only one in which peak excess maternal mortality occurred between November 2020 and February 2021 (135% [40/17]), which coincides not only with the likely emergence, but also with the unprecedented spread of the Gamma variant of concern throughout the North region (especially in Manaus [24]) and the rest of Brazil soon afterwards [25, 26].

The evaluation of gestational risk is age-dependent, as pregnant women aged 35 to 49 years are more prone to systemic complications, especially hypertensive disorders such as preeclampsia and eclampsia, both part of the group of direct causes of maternal death [27, 28]. Throughout the study period, however, the most prominent causes of death in the 35-49-year-old age category were indirect (57.9% [850/1469], not shown in table), in contrast with the

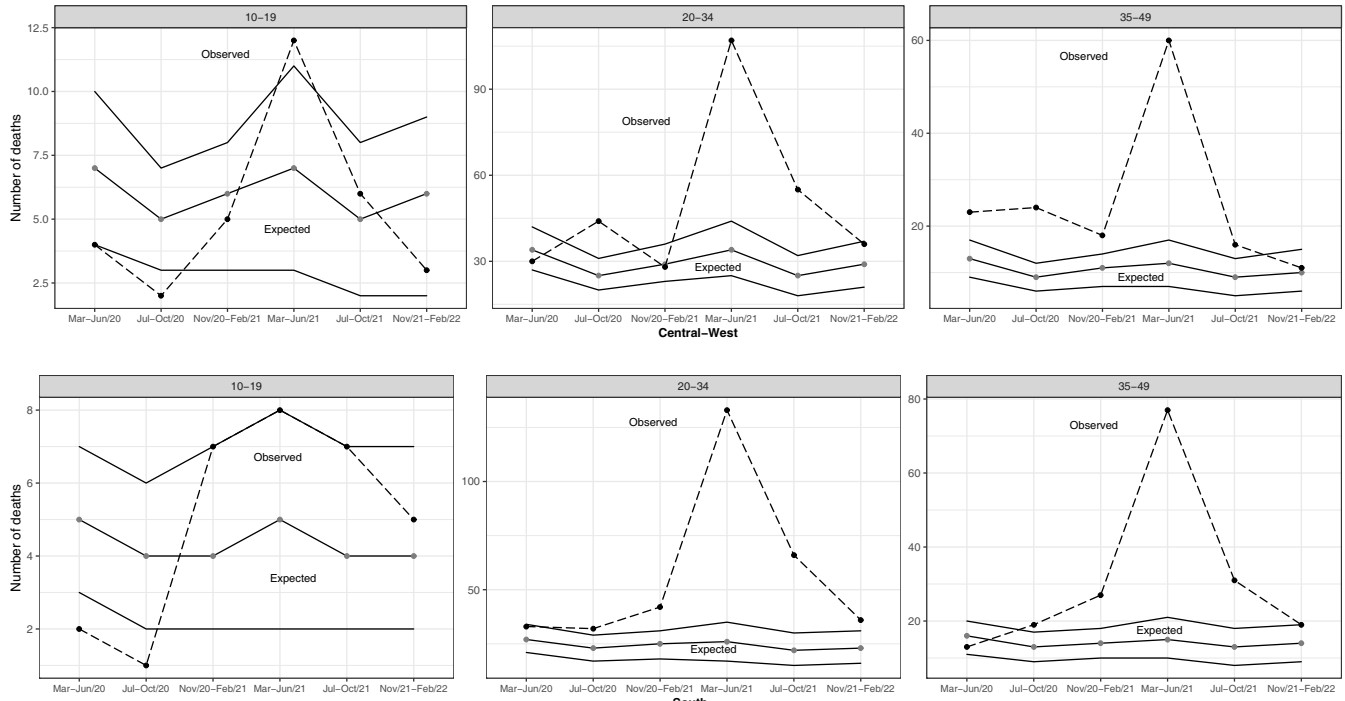

**Fig 3. Observed and expected maternal deaths according to age group and six consecutive four-month periods, Central-West and South regions, 2020 to 2022.**

pattern prior to the COVID-19 epidemic, where direct causes were most responsible for maternal deaths in Brazil [29]. Therefore, the observed reversal in groups of causes may reflect part of the indirect effect of the pandemic on maternal mortality in Brazil. The fact that 42.5% (624/1469, not shown in table) of deaths among women aged 35 to 49 years were attributed to "other viral diseases complicating pregnancy, childbirth, and the puerperium" (O98.5) reinforces this hypothesis, since this atypical pattern may be associated with developing COVID-19.

We stress that the greatest excess in maternal deaths occurred from March to June 2021 among women aged 20–34 and 35–49 in almost every region, with explosive behavior in the South (413% [77/15]) and Central-West (400% [60/12]) among women aged 35–49. This coincided with the time of peak mortality due to COVID-19 at the national level [30]; in early April 2021 about 4,200 individuals died of the disease in a single day [31]. Other studies have encountered substantial maternal deaths due to COVID-19 in the Americas, especially in advanced maternal ages [32]. The occurrence of excess maternal deaths during the critical period of March to June 2021 likely also reflects the late and slow adoption of measures to control and mitigate the epidemic's effects in the country [33–35], such as vaccination of pregnant and postpartum women [2], associated with a substantial protective effect against SARS-CoV-2 infection, severe cases, and deaths [36–39].

From 2018 to 2019, the number of maternal deaths due to direct causes showed a small decrease compared to the two-year period from 2015 to 2016 [7]. Comparing the data from the period from 2020 to 2021 with those from 2018 to 2019, a slight decrease in the number of maternal by direct causes can be observed [7]. With regard to the number of deaths due to indirect causes in Brazil, this number remained relatively stable in the period from 2018 to

**Table 5. Description of the interaction terms between the year of occurrence of death and age group at the regional and national levels, Brazil, 2020 to 2022.**

| North | |
|---|---|
| **Age group (years)** | **p value** |
| 10–19 | 0.873 |
| 20–34 | **0.121** |
| 35–49 | 0.533 |

| Northeast | |
|---|---|
| **Age group (years)** | **p value** |
| 10–19 | **0.021** |
| 20–34 | **0.001** |
| 35–49 | **0.041** |

| Central-West | |
|---|---|
| **Age group (years)** | **p value** |
| 10–19 | 0.667 |
| 20–34 | 0.971 |
| 35–49 | 0.460 |

| South | |
|---|---|
| **Age group (years)** | **p value** |
| 10–19 | 0.651 |
| 20–34 | **0.047** |
| 35–49 | 0.792 |

| Southeast | |
|---|---|
| **Age group (years)** | **p value** |
| 10–19 | 0.945 |
| 20–34 | **0.136** |
| 35–49 | 0.711 |

| Brazil | |
|---|---|
| **Age group (years)** | **p value** |
| 10–19 | **0.164** |
| 20–34 | **0.004** |
| 35–49 | **0.192** |

2019 compared to 2015 to 2016 [7]. However, comparing data from 2020 to 2021 with those from 2018 to 2019, a 187% increase in maternal deaths due to indirect causes can be observed in Brazil [7], not only corroborating our study's findings but also demonstrating the COVID-19 pandemic's disastrous impact on maternal mortality in the country through indirect causes of maternal death.

Compared to a previous analysis on excess maternal mortality in Brazil [2], this study not only confirms fears that the percentage of excess maternal deaths in Year 2 of the COVID-19 epidemic would be even larger compared to Year 1, but also provides documentation of the unexpected negative impact in the South region of the country in Year 2 and of the slight difference in the percentage of excess maternal deaths in the North region between Years 1 and 2. It is also worth highlighting that we presented estimates by age groups that are more easily comparable with the maternal mortality literature, in addition to more accurate predictions in subanalyses with smaller sample sizes, since our minimum temporal aggregation is now four-month rather than three-month periods, as was our initial approach [2].

Thus, our findings suggest that the indirect effects of the failed management of the COVID-19 epidemic substantially altered maternal mortality patterns in the country, regardless of

region and age group. Not only did we present the substantial excess mortality in women and girls aged 10–19, but also the atypical patterns of death among women aged 20–34 and 35–49, predominantly due to indirect causes.

There are some limitations to our study design. Due to the small number of maternal deaths reported in regions as the North and Central-West, it was not possible to disaggregate our estimates of excess deaths monthly or to explore the intersecting effects of racism, socioeconomic inequality, and comorbidities, for example, on maternal mortality in Brazil. The January and February 2022 data are provisional death counts, unlike the other months evaluated during the first two years of the pandemic (the data from March 2020 to December 2021 are considered finalized by the Brazilian Ministry of Health). However, since we analyzed the maternal death counts more than 390 days after the end of our study period, the possibility of underestimation is limited and would have little influence on our current interpretation, specifically for the November 2021 to February 2022 quadrimester. We cannot overlook that the maternal deaths estimates may also be affected by underreporting, due to the deterioration of routine surveillance systems in crisis-affected and resource-poor settings [40], especially in the Brazilian North and Northeast, both strongly impacted by the pandemic and socioeconomically disadvantaged compared to the South and Southeast regions [41].

This study's strengths include the use of the counterfactual approach to estimate the impact of the COVID-19 pandemic on maternal deaths, especially its indirect effects [11] in a context historically characterized by an inequitable distribution of health personnel and facilities, apart from regional time-dependent healthcare pressures, including the infamous Manaus oxygen crisis in January 2021 [24]. We also emphasize that we did not evaluate the pandemic in a single period, but in six consecutive quadrimesters throughout its first two years. Lastly, this approach allowed us to identify similarities and discrepancies in maternal deaths patterns, since this indicator changed substantially according to age-group, region of residence and time-period, likely reflecting the short and long-term consequences of the COVID-19 pandemic on maternal deaths.

## Conclusion

After two pandemic years in Brazil, we observed significantly high maternal excess deaths regardless of region and pandemic year, especially between March and June 2021, the period of peak mortality due to COVID-19. We cannot emphasize enough the importance of a timely and effective epidemic response, not only to prevent tragic, avoidable deaths during acute periods, but also to counter the residual effects of the COVID-19 pandemic and of new crises in a manner that is sensitive to low-and middle-income country contexts, such as Brazil's clear distancing from its commitment through the Sustainable Development Goals (SDGs) to reduce maternal mortality to at most 30 cases per 100 thousand live births by 2030.

## Supporting information

**S1 Table. Description of causes of death among victims of maternal deaths, by each block of chapter XV of the International Statistical Classification of Diseases and Related Health Problems (ICD) 10th Revision, March 2020 to May 2021, Brazil.**
(DOCX)

**S2 Table. Description of the five main causes of death from the blocks containing the most information on maternal causes of death in chapter XV of International Statistical Classification of Diseases and Related Health Problems (ICD) 10th Revision, according six**

consecutive for-month periods, March 2020 to February 2022, Brazil.
(DOCX)

## Author Contributions

**Conceptualization:** Jesem Douglas Yamall Orellana.

**Data curation:** Jesem Douglas Yamall Orellana.

**Formal analysis:** Jesem Douglas Yamall Orellana.

**Methodology:** Jesem Douglas Yamall Orellana.

**Project administration:** Jesem Douglas Yamall Orellana, Daniel Gray Paschoal Leventhal.

**Supervision:** Jesem Douglas Yamall Orellana, Daniel Gray Paschoal Leventhal, María del Pilar Flores-Quispe, Lihsieh Marrero, Nadège Jacques, Lina Sofía Morón-Duarte, Cynthia Boschi-Pinto.

**Visualization:** Jesem Douglas Yamall Orellana, Daniel Gray Paschoal Leventhal, María del Pilar Flores-Quispe, Lihsieh Marrero, Nadège Jacques, Lina Sofía Morón-Duarte, Cynthia Boschi-Pinto.

**Writing – original draft:** Jesem Douglas Yamall Orellana, Daniel Gray Paschoal Leventhal, María del Pilar Flores-Quispe, Lihsieh Marrero, Nadège Jacques, Lina Sofía Morón-Duarte, Cynthia Boschi-Pinto.

**Writing – review & editing:** Jesem Douglas Yamall Orellana, Daniel Gray Paschoal Leventhal, María del Pilar Flores-Quispe, Lihsieh Marrero, Nadège Jacques, Lina Sofía Morón-Duarte, Cynthia Boschi-Pinto.

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
