## [Decision Letter · Decision Letter 0]

15 Nov 2023

PONE-D-23-24297Impact of the COVID-19 pandemic on excess maternal deaths in Brazil: A two-year assessmentPLOS ONE

Dear Dr.  Leventhal

Thank you for submitting your manuscript to PLOS ONE. After careful consideration, we feel that it has merit but does not fully meet PLOS ONE’s publication criteria as it currently stands. Therefore, we invite you to submit a revised version of the manuscript that addresses the points raised during the review process.

ACADEMIC EDITOR: Please insert comments here and delete this placeholder text when finished. Be sure to:

The manuscript is relevant to public health and women's health studies. Below are some suggestions for improvement.

Summary

Highlight which statistical method was used in the analyses and the R libraries.

Introduction

The authors should discuss the contributions of this manuscript with the one previously published in 2022: Orellana J, Jacques N, Leventhal D.G.P., Marrero L, Morón-Duarte LS. Excess maternal mortality in Brazil: Regional inequalities and trajectories during the COVID-19 epidemic. PLoS One. 20 Oct 2022;17(10):e0275333).

Methodology

• Evaluate the data quarterly to dialogue with the manuscript (Orellana J, Jacques N, Leventhal D.G.P., Marrero L, Morón-Duarte LS. Excess maternal mortality in Brazil: Regional inequalities and trajectories during the COVID-19 epidemic. PLoS One Oct 20, 2022;17(10):e0275333).

• Present ICD10 codings in a table to facilitate reading.

• Was any methodology applied to correct deaths? If not, what is the reason?

LAURENTI, R.; MELLO JORGE, M.H.P. of M.; GOTLIEB, S. L. D. Maternal mortality in Brazilian capitals: some characteristics and estimates of an adjustment factor.Rev. Brazilian Epidemiology, São Paulo, v. 4, p. 449-460, Dec. 2004. Available at: http://www.scielo.br/pdf/rbepid/v7n4/08.pdf.

LUIZAGA, C.T.M. et al. Maternal deaths: revision of the correction factor for official data. Epidemiol. Serve. Health, Brasília, v. 1, pg. Mar 8 to 15th. 2010. Available at: http://scielo.iec.gov.br/pdf/ess/v19n1/v19n1a02.pdf. Accessed on: Mar 13, 2018

• Present the formulas of the statistical models used.

Discussion

Highlight the similarities and differences between this article and the previous manuscript (Orellana J, Jacques N, Leventhal D.G.P., Marrero L, Morón-Duarte LS. Excess maternal mortality in Brazil: Regional inequalities and trajectories during the COVID-19 epidemic PLoS One. Oct 20, 2022;17(10):e0275333).

Conclusion

It is too short. Authors need to indicate the contributions of their results to public health assessment and planning. What are the factors that develop into the exposed reality? Moreover, it will make it challenging to achieve the goals of the Sustainable Millennium Goals.

We look forward to receiving your revised manuscript.

Kind regards,

Karina Cardoso Meira, Ph.D

Academic Editor

PLOS ONE

Journal Requirements:

Additional Editor Comments (if provided):

Dear Leventhal

The manuscript is relevant to public health and women's health studies. Below are some suggestions for improvement.

Summary

Highlight which statistical method was used in the analyses and the R libraries.

Introduction

The authors should discuss the contributions of this manuscript with the one previously published in 2022: Orellana J, Jacques N, Leventhal D.G.P., Marrero L, Morón-Duarte LS. Excess maternal mortality in Brazil: Regional inequalities and trajectories during the COVID-19 epidemic. PLoS One. 20 Oct 2022;17(10):e0275333).

Methodology

• Evaluate the data quarterly to dialogue with the manuscript (Orellana J, Jacques N, Leventhal D.G.P., Marrero L, Morón-Duarte LS. Excess maternal mortality in Brazil: Regional inequalities and trajectories during the COVID-19 epidemic. PLoS One Oct 20, 2022;17(10):e0275333).

• Present ICD10 codings in a table to facilitate reading.

• Was any methodology applied to correct deaths? If not, what is the reason?

LAURENTI, R.; MELLO JORGE, M.H.P. of M.; GOTLIEB, S. L. D. Maternal mortality in Brazilian capitals: some characteristics and estimates of an adjustment factor.Rev. Brazilian Epidemiology, São Paulo, v. 4, p. 449-460, Dec. 2004. Available at: http://www.scielo.br/pdf/rbepid/v7n4/08.pdf.

LUIZAGA, C.T.M. et al. Maternal deaths: revision of the correction factor for official data. Epidemiol. Serve. Health, Brasília, v. 1, pg. Mar 8 to 15th. 2010. Available at: http://scielo.iec.gov.br/pdf/ess/v19n1/v19n1a02.pdf. Accessed on: Mar 13, 2018

• Present the formulas of the statistical models used.

Discussion

Highlight the similarities and differences between this article and the previous manuscript (Orellana J, Jacques N, Leventhal D.G.P., Marrero L, Morón-Duarte LS. Excess maternal mortality in Brazil: Regional inequalities and trajectories during the COVID-19 epidemic PLoS One. Oct 20, 2022;17(10):e0275333).

Conclusion

It is too short. Authors need to indicate the contributions of their results to public health assessment and planning. What are the factors that develop into the exposed reality? Moreover, it will make it challenging to achieve the goals of the Sustainable Millennium Goals.

Reviewers' comments:

Reviewer's Responses to Questions

**Comments to the Author**

1. Is the manuscript technically sound, and do the data support the conclusions?

Reviewer #1: Partly

2. Has the statistical analysis been performed appropriately and rigorously? 

Reviewer #1: Yes

3. Have the authors made all data underlying the findings in their manuscript fully available?

Reviewer #1: Yes

4. Is the manuscript presented in an intelligible fashion and written in standard English?

Reviewer #1: No

5. Review Comments to the Author

Reviewer #1: This manuscript is essentially an updated version of an article published a year ago (Orellana J, Jacques N, Leventhal DGP, Marrero L, Morón-Duarte LS. Excess maternal mortality in Brazil: Regional inequalities and trajectories during the COVID-19 epidemic. PLoS One. 2022 Oct 20;17(10):e0275333). Its publication could be justifiable by the fact of adding to the analysis presented in the previous work the period between June 2021 and February 2022, still relevant for the COVID-19 pandemics in Brazil. However, it would be helpful if it kept the same trimestral time division and the same structure of the paper published, in order to “talk” more directly and easily with it. The option for presenting only one table with data for all regions and the country e much better than the option chosen in this version. The organization of the results’ text by also sounds better in the previous article.

I would recommend that the authors make more explicit the connection of the work with the previous one, underlining why the updated version was done. Perhaps some background and methodological elements may be synthesized, referring that more information is in the other article.

If both articles are similar, it better that their similarities and complementarities are clear.

The text could be reviewed to become more fluid; some style repetition makes it a little boring. Small details such as using “from 2018-2019” should also be avoided. The authors may chose between “from 2018 to 2019” or “in (the period) 2018-2019.

6. PLOS authors have the option to publish the peer review history of their article (what does this mean?). If published, this will include your full peer review and any attached files.

Reviewer #1: No

---

## [Author Response · Author response to Decision Letter 0]

27 Nov 2023

Summary

Highlight which statistical method was used in the analyses and the R libraries.

RESPONSE: We appreciate your recommendation; information pertaining to the packages used during statistical modeling was added to the abstract, as follows: “Analyses were performed in R version 4.1.2, RStudio, version 2023.03.1+446 and carried out with support from the “mgcv” and “plot_model” libraries.” This change is also reflected in the Methods section (lines 231-233).

Introduction

The authors should discuss the contributions of this manuscript with the one previously published in 2022: Orellana J, Jacques N, Leventhal D.G.P., Marrero L, Morón-Duarte LS. Excess maternal mortality in Brazil: Regional inequalities and trajectories during the COVID-19 epidemic. PLoS One. 20 Oct 2022;17(10):e0275333).

RESPONSE: We appreciate the suggestion and agree that the Introduction can be improved by highlighting these analyses’ contribution in relation to the previous article on excess maternal mortality in Brazil. We added the following text to the Introduction section: “A previous study examining data from March 2020 to May 2021 estimated a 70% excess in maternal mortality in Brazil, with wide variations by region and time period evaluated [2]. However, the study did not allow for an evaluation of the possible indirect effects of the COVID-19 epidemic on maternal mortality in traditional age groupings (10-19 years, 20-34 years, and 35-49 years). In addition, the study did not assess excess maternal deaths in the second and most critical year of the epidemic in different regions of Brazil, especially in the months following the late, uninterrupted distribution of vaccines against COVID-19 to pregnant and post-partum women beginning in July 2021.”

Methodology

• Evaluate the data quarterly to dialogue with the manuscript (Orellana J, Jacques N, Leventhal D.G.P., Marrero L, Morón-Duarte LS. Excess maternal mortality in Brazil: Regional inequalities and trajectories during the COVID-19 epidemic. PLoS One Oct 20, 2022;17(10):e0275333).

RESPONSE: We thank you for your pertinent comment. Nevertheless, we have chosen to maintain the analysis by quadrimester because 1) the overall estimates from Years 1 and 2 will be the same both in analyses by trimester and by quadrimester, and, in particular, 2) the analysis by quadrimester improves the accuracy of our predictions (especially those related to maternal deaths in women aged 10-19 years, in regions or even in other subanalyses within regions with a smaller “n,” like the Central-West and South regions) and, finally, 3) the quadrimester analysis simplifies the presentation and discussion of the results of the two-year assessment. The rationale for this decision is provided in the Methods section, as follows: “The chosen level of aggregation (by quadrimester) was agreed upon with the aim to improve the accuracy of our predictions, especially those pertaining to maternal deaths in women aged 10-19 years or even in subanalyses in regions with smaller samples, like the Central-West and South regions.”

Present ICD10 codings in a table to facilitate reading.

RESPONSE: Thank you and we agree with your suggestion to include a table, substituting the previous description in the text. The table is entitled “Table 1. Maternal deaths according to codings from the International Statistical Classification of Diseases and Related Health Problems, 10th Revision (ICD-10)”.

Table 1. Maternal deaths according to codings from the International Statistical Classification of Diseases and Related Health Problems, 10th Revision (ICD-10)”.

ICD-10 Codings Group of maternal deaths

O00-O99 Pregnancy, childbirth and the puerperium*

B20-B24 Human immunodeficiency virus [HIV] disease**

D39.2 malignant or invasive hydatidiform mole**

E23.0 Hypopituitarism**

M83.0 Puerperal osteomalácia***

A34 Obstetrical tetanus***

F53 Mental and behavioural disorders associated with the puerperium, not elsewhere classified***

*Except deaths outside of the 42-day postpartum period (codes O96 and O97)

**With a gestational report at the time of death or up to 42 days before death

***With death occurring up to 42 days after the termination of pregnancy or when there was no information on the time elapsed between the termination of gestation and death

• Was any methodology applied to correct deaths? If not, what is the reason?

LAURENTI, R.; MELLO JORGE, M.H.P. of M.; GOTLIEB, S. L. D. Maternal mortality in Brazilian capitals: some characteristics and estimates of an adjustment factor.Rev. Brazilian Epidemiology, São Paulo, v. 4, p. 449-460, Dec. 2004. Available at: http://www.scielo.br/pdf/rbepid/v7n4/08.pdf.

LUIZAGA, C.T.M. et al. Maternal deaths: revision of the correction factor for official data. Epidemiol. Serve. Health, Brasília, v. 1, pg. Mar 8 to 15th. 2010. Available at: http://scielo.iec.gov.br/pdf/ess/v19n1/v19n1a02.pdf. Accessed on: Mar 13, 2018

RESPONSE: We appreciate your important suggestion and for the purposes of clarification have included the reason for not using the method of correcting maternal deaths in the revised version of the Methods section. The altered text is as follows: “We analyzed data from March 2020 to February 2022. We did not apply correction factors to the maternal mortality estimates, as the focus of the article is to present and discuss excess deaths as a percentage. These do not change, with or without a fixed correction on the observed or estimated counts.”

• Present the formulas of the statistical models used.

RESPONSE: We thank you for your important suggestion. We have included our formula with a textual explanation of each variable used in the Methods section (lines 187-213).

Discussion

Highlight the similarities and differences between this article and the previous manuscript (Orellana J, Jacques N, Leventhal D.G.P., Marrero L, Morón-Duarte LS. Excess maternal mortality in Brazil: Regional inequalities and trajectories during the COVID-19 epidemic PLoS One. Oct 20, 2022;17(10):e0275333).

RESPONSE: We are grateful for and agree on your timely suggestion. For this reason, we have added to the Discussion section, highlighting the contributions of the current version of the article assessing excess maternal mortality in Brazil, as follows: “Compared to a previous analysis on excess maternal mortality in Brazil [2], this study not only confirms fears that the percentage of excess maternal deaths in Year 2 of the COVID-19 epidemic would be even larger compared to Year 1, but also provides documentation of the unexpected negative impact in the South region of the country in Year 2 and of the slight difference in the percentage of excess maternal deaths in the North region between Years 1 and 2. It is also worth highlighting that we presented estimates by age groups that are more easily comparable with the maternal mortality literature, in addition to more accurate predictions in subanalyses with smaller sample sizes, since our minimum temporal aggregation is now four-month rather than three-month periods, as was our initial approach [2].”

Conclusion

It is too short. Authors need to indicate the contributions of their results to public health assessment and planning. What are the factors that develop into the exposed reality? Moreover, it will make it challenging to achieve the goals of the Sustainable Millennium Goals.

RESPONSE: We thank you for and agree with this crucial suggestion. For this reason, we have added the following text: “We cannot emphasize enough the importance of a timely and effective epidemic response, not only to prevent tragic, avoidable deaths during acute periods, but also to counter the residual effects of the COVID-19 pandemic and of new crises in a manner that is sensitive to low-and middle-income country contexts, such as Brazil’s clear distancing from its commitment through the Sustainable Development Goals (SDGs) to reduce maternal mortality to at most 30 cases per 100 thousand live births by 2030.”

Reviewer #1: 

This manuscript is essentially an updated version of an article published a year ago (Orellana J, Jacques N, Leventhal DGP, Marrero L, Morón-Duarte LS. Excess maternal mortality in Brazil: Regional inequalities and trajectories during the COVID-19 epidemic. PLoS One. 2022 Oct 20;17(10):e0275333). Its publication could be justifiable by the fact of adding to the analysis presented in the previous work the period between June 2021 and February 2022, still relevant for the COVID-19 pandemics in Brazil. However, it would be helpful if it kept the same trimestral time division and the same structure of the paper published, in order to “talk” more directly and easily with it.

RESPONSE: Thank you for your suggestions. Please see our responses to the academic editor’s suggestions dealing with the Introduction and Methods sections. 

The option for presenting only one table with data for all regions and the country e much better than the option chosen in this version. 

RESPONSE: Thank you for the suggestion. However, the results from our second article would be disproportionally represented in one table, as we are dealing with data from Years 1 and 2 as well as their overall distribution for Brazil and all its regions, unlike in our first article. For this reason, we have decided to present results in more than one table. 

The organization of the results’ text by also sounds better in the previous article.

RESPONSE: We appreciate the comment, but we have chosen to maintain the current organization of the results, due to the larger volume of data being presented. 

I would recommend that the authors make more explicit the connection of the work with the previous one, underlining why the updated version was done. 

RESPONSE: Thank you for your suggestion. Please see our responses to the academic editor for the Introduction and Discussion sections. 

Perhaps some background and methodological elements may be synthesized, referring that more information is in the other article.

RESPONSE: We appreciate the recommendation. We have opted to provide more clarity to certain parts of the Methods section as well as to synthesize part of the text in a table. 

If both articles are similar, it better that their similarities and complementarities are clear.

RESPONSE: Thank you for your suggestion. Please see our reply to your penultimate suggestion. 

The text could be reviewed to become more fluid; some style repetition makes it a little boring.

RESPONSE: We thank you for your constructive comment and believe that the revised version of the present article has made the text clearer and more fluid. 

Small details such as using “from 2018-2019” should also be avoided. 

RESPONSE: Thank you for the constructive commentary. We have made the appropriate changes.

The authors may chose between “from 2018 to 2019” or “in (the period) 2018-2019.

RESPONSE: Thank you for your helpful suggestion. We have changed “2018-2019” to “2018 to 2019.”

---

## [Decision Letter · Decision Letter 1]

17 Jan 2024

PONE-D-23-24297R1Impact of the COVID-19 pandemic on excess maternal deaths in Brazil: A two-year assessmentPLOS ONE

Dear Dr. Leventhal,

Thank you for submitting your manuscript to PLOS ONE. After careful consideration, we feel that it has merit but does not fully meet PLOS ONE’s publication criteria as it currently stands. Therefore, we invite you to submit a revised version of the manuscript that addresses the points raised during the review process.

 The manuscript requires minor adjustments to the table titles and the model formula, specifically incorporating the random error, which is currently missing.

We look forward to receiving your revised manuscript.

Kind regards,

Karina Cardoso Meira, Ph.D

Academic Editor

PLOS ONE

Journal Requirements:

Reviewers' comments:

Reviewer's Responses to Questions

**Comments to the Author**

1. If the authors have adequately addressed your comments raised in a previous round of review and you feel that this manuscript is now acceptable for publication, you may indicate that here to bypass the “Comments to the Author” section, enter your conflict of interest statement in the “Confidential to Editor” section, and submit your "Accept" recommendation.

Reviewer #1: All comments have been addressed

2. Is the manuscript technically sound, and do the data support the conclusions?

Reviewer #1: Yes

3. Has the statistical analysis been performed appropriately and rigorously? 

Reviewer #1: Yes

4. Have the authors made all data underlying the findings in their manuscript fully available?

Reviewer #1: Yes

5. Is the manuscript presented in an intelligible fashion and written in standard English?

Reviewer #1: Yes

6. Review Comments to the Author

Reviewer #1: The questions raised previously were satisfactorily addressed. My only note at this point is that the titles of Table 2, 3 and 4 need to be appropriately defined.

7. PLOS authors have the option to publish the peer review history of their article (what does this mean?). If published, this will include your full peer review and any attached files.

Reviewer #1: No

---

## [Author Response · Author response to Decision Letter 1]

21 Jan 2024

ACADEMIC EDITOR:

The manuscript requires minor adjustments to the table titles and the model formula, specifically incorporating the random error, which is currently missing.

We thank the editor for their comments. Regarding the subject of the formula we provided, we have opted to clarify a few points about the quasipoisson generalized additive model (GAM) instead of including the random error, as follows in lines 155-161: “The mean parameter θ specifically is used to address overdispersion (when the variance of count data is larger than the mean) and does not have a specific associated distribution, acting as a scaling factor that adjusts the variance to reflect the observed overdispersion. Therefore, the modeled distribution is proportional to the mean estimated by the model, expressed mathematically as Var(Y∣X)= ø ⋅ µ, where µ is the expected mean. Furthermore, when the value of ø is larger than one, this indicates that the data display more variability than expected (assuming a Poisson distribution in which both the mean and the variance are equal).”

We have also changed the title of tables 2-4 to reflect the information contained therein (e.g., Table 2. Observed and expected maternal deaths, Brazil and Southeast region, 2020-2022).

Reviewer #1:

The questions raised previously were satisfactorily addressed. My only note at this point is that the titles of Table 2, 3 and 4 need to be appropriately defined.

We are grateful for the attention given to this matter. Accordingly, we have named Tables 2-4 based on the information they present, as mentioned above in the response to the editor.

---

## [Editor Report · Decision Letter 2]

31 Jan 2024

Impact of the COVID-19 pandemic on excess maternal deaths in Brazil: A two-year assessment

PONE-D-23-24297R2

Dear Dr. Leventhal,

We’re pleased to inform you that your manuscript has been judged scientifically suitable for publication and will be formally accepted for publication once it meets all outstanding technical requirements.

Kind regards,

Karina Cardoso Meira, Ph.D

Academic Editor

PLOS ONE
---

## [Editor Report · Acceptance letter]

18 Mar 2024

PONE-D-23-24297R2 

PLOS ONE

Dear Dr. Leventhal, 

I'm pleased to inform you that your manuscript has been deemed suitable for publication in PLOS ONE. Congratulations! Your manuscript is now being handed over to our production team.

Kind regards, 

on behalf of

Dr. Karina Cardoso Meira 

Academic Editor

PLOS ONE